# Bagasse Cellulose Composite Superabsorbent Material with Double-Crosslinking Network Using Chemical Modified Nano-CaCO_3_ Reinforcing Strategy

**DOI:** 10.3390/nano12091459

**Published:** 2022-04-25

**Authors:** Xinling Xie, Li Ma, Yongmei Chen, Xuan Luo, Minggui Long, Hongbing Ji, Jianhua Chen

**Affiliations:** 1Guangxi Key Laboratory of Petrochemical Resource Processing and Process Intensification Technology, School of Chemistry and Chemical Engineering, Guangxi University, Nanning 530004, China; mali15956442057@163.com (L.M.); luoxuan@gxu.edu.cn (X.L.); jihb@mail.sysu.edu.cn (H.J.); 2Guilin Zhuorui Food Ingredients Co., Ltd., Guilin 541001, China; lilychym-123@163.com (Y.C.); longmg@126.com (M.L.); 3Fine Chemical Industry Research Institute, School of Chemistry, Sun Yat-sen University, Guangzhou 510275, China; 4School of Resources, Environment, and Materials, Guangxi University, Nanning 530004, China; jhchen@gxu.edu.cn

**Keywords:** bagasse cellulose, nano-CaCO_3_, double crosslinked network structure, composite superabsorbent

## Abstract

To improve the salt resistance of superabsorbent materials and the gel strength of superabsorbent materials after water absorption, a bagasse cellulose-based network structure composite superabsorbent (CAAMC) was prepared via graft copolymerization of acrylamide/acrylic acid (AM/AA) onto bagasse cellulose using silane coupling agent modified nano-CaCO_3_ (MNC) and N,N′-methylene bisacrylamide (MBA) as a double crosslinker. The acrylamide/acrylic acid was chemically crosslinked with modified nano-CaCO_3_ by C-N, and a stable double crosslinked (DC) network CAAMC was formed under the joint crosslinking of N,N′-methylene bisacrylamide and modified nano-CaCO_3_. Modified nano-CaCO_3_ plays a dual role of crosslinking agent and the filler, and the gel strength of composite superabsorbent is two times higher than that of N,N′-methylene bisacrylamide single crosslinking. The maximum absorbency of CAAMC reached 712 g/g for deionized water and 72 g/g for 0.9 wt% NaCl solution. The adsorption process of CAAMC was simulated by materials studio, and the maximum adsorption energy of amino and carboxyl groups for water molecules is −2.413 kJ/mol and −2.240 kJ/mol, respectively. According to the results of CAAMC soil water retention, a small amount of CAAMC can greatly improve the soil water retention effect.

## 1. Introduction

Superabsorbent resin (SAR) is a kind of polymer with a crosslinked three-dimensional network structure; it can absorb hundreds or even thousands of times its weight in water, and it can hold water under specific pressures without separation [1,2], which makes it widely used in many fields [3,4,5,6]. However, in the past, the raw materials of SAR for commercial applications mainly come from poorly degradable petrochemical products such as polyacrylamide and polyacrylic acid [7,8] and would cause severe environmental pollution. In addition, some SAR show poor salt absorption of only 40–50 g/g and lower gel strength after water absorption [9,10,11], and it is difficult to increase water absorbency and gel strength after absorption simultaneously [12]. The application of SAR in personal hygiene products and soil water-holding agents was limited. Therefore, degradable biomass-based composite superabsorbent resin with different functional groups has attracted the attraction of researchers.

Cellulose is the most abundant natural polymer and contains a large number of hydroxyl groups that can be used to alter its properties chemically. Because of its excellent properties, such as biodegradability, biocompatibility, and environmental friendliness, cellulose has become an ideal backbone for synthesizing biodegradable superabsorbent polymers [13,14]. Bagasse, a by-product of the sugar cane industry containing 40–50 wt% cellulose, 25–35 wt% hemicellulose, and 18–24 wt% lignin, contains many substances of hydroxyl and phenolic groups [15,16,17,18,19,20,21]. According to a survey, global production of sugarcane in recent years was about 260 million tons/year [22], which would produce a huge amount of bagasse. A small part of this bagasse was used to make paper, and the rest was burnt as a means of solid waste disposal [23], which wasted resources and seriously polluted the environment [24]. The application of bagasse cellulose-based SAR enriched the raw materials of SAR, and bagasse-based biodegradable SAR increases the added value of bagasse and reduces environmental pollution [25,26,27,28]. According to literature reports, the current superabsorbent resin based on bagasse cellulose can be naturally degraded and has a good absorbency in deionized water. However, weak gel strength after water absorption and low salt tolerance are still the problems faced by bagasse cellulose-based SARs.

At present, the superabsorbent resin modified by natural polymers has partial biodegradability, but its salt resistance and gel strength after water absorption are relatively poor. To improve the salt tolerance and gel strength after water absorption of bagasse-based SAR, inorganic nanoparticles are added into bagasse-based SAR [29]. CaCO_3_ is cheap and less sensitive to salt ions natural mineral, which is considered an ideal modifier of SAR. However, the small size effect of CaCO_3_ nanoparticles easily unites and disperses unevenly when they are directly dispersed into the natural polymer components, which leads to obvious interface defects in the organic–inorganic components and brittle fracture after water absorption [30,31]. To solve this problem, some researchers have modified the surface of CaCO_3_ so that it can combine with organic matter through chemical bonds [32].

The swelling kinetics and mechanism of simulating the swelling process play an important guide in the construction of superabsorbent resins. However, most of the literature is based on water adsorption kinetic models, i.e., Fickian water diffusion mechanism and pseudo-first/second-order swelling kinetic model were used to illustrate the macro water absorption process of SAR, and it can be difficult to clearly visualize the structural changes during the water absorption process [33,34]. Additionally, the specific roles of the functional groups contained in the different SAR are not described.

With the above consideration, the bagasse cellulose was used as the raw material, nano-CaCO_3_ modified with 3-aminopropyltriethoxysilane, and then combined with MBA as a double crosslinking. After the chemical crosslinking reaction, a DC network structure bagasse cellulose-based composites superabsorbent resin (CAAMC) via graft copolymerization was prepared in an aqueous solution. CAAMC was characterized by Fourier transform infrared spectroscopy (FT-IR), X-ray powder diffraction (XRD), scanning electron microscopy (SEM), Energy Dispersive Spectrometer (EDS), contact angle measuring instrument (Sdc-350), X-ray photoelectron spectroscopy (XPS), Thermogravimetric analysis (TG-DTG), and Solid state NMR (^13^C NMR). The effects of the modified nano-CaCO_3_ amount on the water absorbency, water retention, and reswelling performance of the composites were investigated. Our results indicate that the structure of CAAMC can be changed by adjusting the content of modified nano-CaCO_3_, thus significantly improving the mechanical properties and salt tolerance of CAAMC. In addition, Materials Studio was used to calculate the adsorption energy of different functional groups on water molecules and simulate the structural changes of SAR throughout the water absorption process.

## 2. Materials and Methods

### 2.1. Materials

Bagasse was purchased from Guangxi, China (bagasse cellulose content of 80.62 wt%; water content of 10 wt% after pre-treatment). CaCl_2_ and (NH_4_)_2_CO_3_ purchased from Chengdu Kelong Chemical Co., Ltd. (Chengdu, Sichuan Province, China), both of which are of analytical grade, were utilized as received with no further purification. Ethanolamine and triethanolamine were purchased from Xilong Science Co., Ltd. (Shantou, Guangdong Province, China), both of which are of analytical grade. 3-aminopropyltriethoxysilane, potassium persulfate (APS), acrylamide (AM), and N,N′-methylene bisacrylamide (MBA) were purchased from Macklin (Shanghai, China), and all chemical reagents utilized in this study were of analytical grade and were utilized as received with no further purification.

#### 2.1.1. Preparation Method of CAAMC Graft Copolymer

The preparation process of composite SAR CAAMC is shown in Figure 1. The bagasse recovered from the sugar mill was crushed by a grinder and passed through a 60-mesh sieve, which is further extracted based on the procedures in published work [35] to obtain the bagasse cellulose (BC) with abundant hydroxyl groups as the base material, and the morphology (as shown in Appendix A) and chemical structure (as shown in Appendix A) of BC have been characterized.

Nano-CaCO_3_ was prepared by a solid-state reaction. CaCl_2_ (1 g), (NH_4_)_2_CO_3_ (1 g), ethanolamine (0.02 g), and triethanolamine (0.01g) were put into a planetary ball mill tank, milled for 50 min, thoroughly washed, and dried at 60 °C, and nanometer-sized CaCO_3_ could be obtained. Then, 2.0 g of nano-CaCO_3_ and 2.0 g of 3-aminopropyltriethoxysilane (KH550) were weighed into a flask, and 100 mL 95% ethanol was added. After full stirring for 6.5 h, the precipitate was washed with ethanol and dried in an oven at 100 °C for 5 h to obtain modified nano-CaCO_3_ (MNC) (The N_2_ adsorption isotherms (nano-CaCO_3_ (A) and MNC (B)) and pore size distributions (nano-CaCO_3_ (C) and MNC (D) are shown in Appendix A) [36].

Bagasse cellulose (0.75 g) and MNC (0.15 g) were added to 40 mL of deionized water and gelatinized at 90 °C for 40 min in a nitrogen atmosphere. Potassium persulfate (0.12 g) was added after cooling to form hydroxyl radicals. Then, the mixed solution, with a neutralization degree of AA (5 g) 60%, acrylamide (2.5 g), and N, N′-methylene bisacrylamide (0.025 g), was slowly added to a three-neck flask and then reacted at 60 °C for 1.5 h. After the reaction, the samples were dried in an oven at 60 °C. The dried samples were extracted in a Soxhlet extractor with acetone and ethanol for 12 h and then dried in an oven at 60 °C. Finally, the dried samples were ground, crushed, and passed through a 60-mesh sieve for further analysis [37].

#### 2.1.2. Characterization Method

Morphologies of the CAAMC, nano-CaCO_3_, and MNC were examined using a scanning electron microscope (SU8220, Hitachi, Tokyo, Japan) at a voltage of 10 kV. The dried specimens were treated by gold-sputtering to supply the right surface conduction. Energy Dispersive Spectrometer (EDS, SU8220, Hitachi, Tokyo, Japan) analysis was used to detect the elemental composition at a voltage of 15 kV. With potassium bromide as the background, the surface groups of cellulose, CaCO_3_, modified CaCO_3_, and CAAMC were analyzed using Fourier transform infrared spectra (FT-IR, Nicolet FTIRIS10, Thermo Fisher Scientific, Waltham, MA, USA). The surface element composition and valence of cellulose, MNC, and CAAMC were analyzed by X-ray photoelectron spectroscopy (XPS, ESCALAB250, Thermo Fisher Scientific, Waltham, MA, USA) and C1s (284.6 eV) calibration [38]. The contact angles between water droplets with bagasse, BC, and CAAMC were tested using a contact angle measuring instrument (Sdc-350, Dongguan, Guangdong Province, China, 3 μ/d). The particle size and particle size distribution of unmodified nano-CaCO_3_ and MNC were analyzed by a nanoparticle size analyzer (Zetasizer Nano ZS90, Malvern Instruments Co., Ltd., Malvern, UK); thermogravimetric (TG-DTG) analysis was performed on a STA449F3 synchronous thermal analyzer (Netzsch-Geratebau GmbH, Selb, Germany) at a heating rate of 20 °C min^−1^ under N_2_ atmosphere. [39]. X-ray diffraction (XRD) patterns of nano-CaCO_3_, MNC, Cellulose, and CAAMC superabsorbent were also obtained using a diffractometer (SMARTLAB3KW, Rigaku Corporation, Tokyo, Japan) equipped with a Cu Kα radiation source in a scattering angle range from 10° to 80° [29]. The compressive strength of CAAMC saturated with water absorption was tested using a universal electronic material testing machine (Instron Company, Boston, MA, USA); at least 3 measurements were carried out for each sample and the mean value was obtained [40]. Solid-state-^13^C CP/MAS NMR spectra were taken on a Brucker 400M spectrometer with a time domain size of 2048 and 1000 scans.

### 2.2. Measurement of Water Absorbency

The swelling properties of CAAMC in deionized water and 0.9 wt% NaCl solution were tested by the gravimetric method. The dry sample (0.01 g) before weighing was immersed in excess deionized water (250 mL), soaked at room temperature, allowed to reach swelling equilibrium, passed through a 60-mesh sieve, and filtered out the saturated resin. The weight was determined after water absorption again, and the above steps were repeated in other solutions. Three measurements were repeated for each sample [41]. The water absorbency was calculated by Equation (1).
(1)$$Q_{eq}=\frac{m_{2}-m_{1}}{m_{1}}$$
where the water absorbency is *Q_eq_* per gram, g/g, *m*_1_ is the weight of the dried resin, g, and *m*_2_ is the weight of the resin when it reaches swelling equilibrium, g.

### 2.3. Measurement of Water Retention in Soil

In this work, CAAMC of a different quality was added into soil to investigate the water retention ability in soil. The properties were measured at different treatments: control, 50 g of dry soil without CAAMC; 50 g of dry soil mixed well with 0.25 g of CAAMC; 50 g of dry soil mixed well with 0.5 g of CAAMC; 50 g of dry soil mixed well with 0.75 g of samples; 50 g of dry soil mixed well with 1.0 g of CAAMC. Each sample was placed in a plastic cup and weighed (*W*_0_). Then, the mixtures were slowly drenched with tap water and the tube was weighed again (*W*_1_). The beakers were kept at 25 °C and weighed every day (*W_i_*). The water retention (WR, (%)) of soil was calculated by Equation (2) [42].
(2)$$\mathrm{WR}(\%)=\frac{W_{i}-W_{0}}{W_{1}-W_{0}}\times 100\%\;$$

### 2.4. Computation Details

For geometric optimization using the Dmol3 module in Materials Studio, the exchange-dependent energy functional was based on the generalized gradient approximation (m-GGA-M06-L) of Perdew-Burke-Ernzerho’s kinetic energy, and the polarization function (DNF) basis set adopted a double numerical mass basis [43]. The M-GGA-M06-L function can describe the binding energy of hydrogen bonds [44]. Materials Studio was used to simulate the water absorption process, and the adsorption energy (Δ*E*) of functional group amines and carboxyl groups for water molecules in the absorbent resin was calculated using Equation (3).
(3)$$\Delta E=E\mathrm{system}-\left(E_{\mathrm{CAAMC}}+E_{\mathrm{Water}}\right)$$
where *E*_system_ is the total energy of functional groups after absorbing water, kJ/mol; *E*_CAAMC_ is the total energy of unabsorbed water, kJ/mol; and *E*_Water_ is the energy of free water molecules, kJ/mol.

## 3. Results

### 3.1. Characterization of CAAMC

Figure 2A shows the FT-IR spectra of bagasse cellulose, nano-CaCO_3_, MNC, and CAAMC. Compared with unmodified nano-CaCO_3_, modified nano-CaCO_3_ exhibits a characteristic Si-O-CaCO_3_ absorption peak at 1164 and 1051 cm^−1^ [45], and the peak width of the high absorption peak of C-O at 1460 cm^−1^ covers the N-H absorption of the silane coupling agent KH-550 (1584 and 1383 cm^−1^). Moreover, at 2928 cm^−1^ and 2878 cm^−1^, the weak characteristic peak of -CH_2_- appears, and more significantly, the intensity of MNC’s OH- stretching vibration peak noticeably decreases, indicating that KH-550 was hydrolyzed in an aqueous alcohol solution, ethoxy was converted into hydroxyl, and hydroxyl adsorbed on the surface of nano-CaCO_3_ was dehydrated in the solvent to form a chemical connection, a long chain of -Si-O-CaCO_3_, allowing KH-550 to form a flexible organic structure on the nano-CaCO_3_ surface, which also confirmed the chemical bond between KH-550 and nano-CaCO_3_ [46]. The FT-IR spectra of CAAMC shows that what originally belonged to the cellulose -OH peak was weakened, and the peaks at 1401 and 1458 cm^−1^ were the characteristic absorption peaks of -COOH, indicating that acrylic acid has been successfully grafted onto cellulose, and -COOH becomes the main functional group of the superabsorbent polymer. The absorption peak at 1330 cm^−1^ was the absorption peak of C-N, and a peak at 1667 cm^−1^ attributed to C=O stretching vibration. The appearance of these two peaks indicates that acrylamide was successfully grafted onto cellulose skeleton [37,47,48]. The characteristic peak of Si-O appeared at 1169 cm^−1^, indicating that the bagasse/AM/AA copolymer was chemically crosslinked with MNC by C-N, and MNC was involved in the grafting reaction [47,49].

Figure 2B shows the XRD patterns of bagasse cellulose, nano-CaCO_3_, MNC, and CAAMC (The XRD of a single CAAMC is shown in Appendix A). The peak shapes of nano-CaCO_3_ were the same before and after modification, and they all had a triangular structure (space Group R-3c) (JCPDS card number 81–2027). Their primary characteristic peaks are 2θ = 23.1°, 29.5°, 36°, 39.5°, 43.1°, 47.6°, 48.6°, 56.4°, and 57.5°, corresponding to (012), (104), (110), (113), (202), (018), (116), (211), and (122), respectively, which is a typical calcite structure. Furthermore, MNC has a stronger reflection peak than the unmodified nano-CaCO_3_ at the strongest crystal reflector position, 2θ = 29.5°, because of the direct coupling of the coupling agent itself after successful modification of the coupling agent, leading to the growth of the nano-CaCO_3_ crystal surface [45,50]. After treatment, bagasse cellulose belonged to cellulose type I, and the lower crystal reflection of CAAMC disappeared at 2θ = 18.16° and broadened at 22.36°, indicating the destruction of the ordered cellulose structure during polymerization. Acrylamide and acrylic acid were successfully grafted into the primary cellulose chain, consistent with the FT-IR results. Furthermore, the characteristic peak of MNC disappeared because MNC was coated with organic components, indicating that it was involved in graft polymerization and resin network crosslinking processes [51].

Figure 2C,D show the TG-DTG curves of nano-CaCO_3_. As shown in (A) in Figure 2, Figure 3, Figure 4, Figure 5 and Figure 6, the weight loss of unmodified nano-CaCO_3_ mainly occurred at around 700 °C, and the weight loss rate reached a maximum at 727 °C, which was attributed to the thermal decomposition of nano-CaCO_3_ itself. As shown in (B) in Figure 2, Figure 3, Figure 4, Figure 5 and Figure 6, the weight loss of the modified nano-CaCO_3_ (MNC) obtained by silane coupling agent modification mainly occurs at about 700 °C, but the weight loss rate reaches the maximum value at 770 °C, which is obviously higher than that of unmodified nano-CaCO_3_, which indicates the stronger stability of MNC. In addition, comparing the total weight loss of nano-CaCO_3_ and MNC, we found that the weight loss of MNC is 1.49% higher than that of nano-CaCO_3_, which indicates that the modified nano-CaCO_3_ surface has the composition of silane coupling agent, nano-CaCO_3_ having been successfully modified.

For the thermal decomposition of bagasse cellulose (BC), it can be roughly divided into two stages (as shown in (E) in Figure 2, Figure 3, Figure 4, Figure 5 and Figure 6), the first stage weight loss at around 109 °C was attributed to the evaporation of water; the weight of this loss is about 10 wt%. The second stage was the main stage of pyrolysis; the main temperature range is around 200–500 °C, which was mainly produced by the thermal decomposition of bagasse cellulose. The weight loss in this period is 70 wt%, and the remaining mass of about 20 wt% was residual ash [52].

(F) in Figure 2, Figure 3, Figure 4, Figure 5 and Figure 6 shows the TG and DTG curves of CAAMC. The weight loss of CAAMC at 100–300 °C was due to the evaporation of water in the sample [53], the weight loss in the range of 300–550 °C was mainly due to the decomposition of CAAMC, and the weight loss above 700 °C was due to the decomposition of modified calcium carbonate (MNC) [54]. The DTG curve of CAAMC shows that the first weight loss of the sample peaks at 240 °C is due to the evaporation of water. During CAAMC decomposition, DTG exhibited two peaks in weight loss rate, which were caused by the decomposition of carboxyl and amido groups of acrylic acid and acrylamide grafted on cellulose, reaching peaks at 334 °C and 445 °C, respectively [55]. From the TG-DTG results of CAAMC, it can be seen that CAAMC has good thermal stability.

Figure 3A shows that there is a Si-O peak at 531 eV from the O1s peak in MNC, indicating the formation of new Ca-O-Si bonds on the surface of nano-CaCO_3_. In addition, new C-O and C-C bonds can be seen in Figure 3B, and Si 2p peaks can be seen in Figure 3C, indicating that the nano-CaCO_3_ surface is coated with silane coupling agent. FT-IR characterization also confirmed this result, thus confirming the successful modification of nano-CaCO_3_ [56].

Figure 4 shows the XPS of CAAMC. The XPS spectrum shows that the O1s peak is divided into four peaks, C=O, C-O, O-C=O, and CO_3_^2−^ (as shown in Figure 4A), with different binding energies. CO_3_^2^^−^ is the emerging peak compared with the XPS spectrum of the superabsorbent resin without MNC, indicating that the SAR was successfully compounded with MNC, and the peak position of CO_3_^2−^ contained in CAAMC is shifted from that of CO_3_^2−^ contained in MNC, indicating that MNC is involved in the grafting reaction. Moreover, Figure 4B shows a Si 2p peak, indicating the successful grafting of MNC and the resin. Figure 4C indicates that the C 1s peak is divided into five peaks, which were assigned to O-C=O, C-N, C=O, C-O, and C-C bonds, having different binding energies. Both C=O and C-N bonds were obtained from the carbonyl and amino groups, respectively. Figure 4D shows an N 1s peak, and the peak is attributed to N-H, which indicates the successful grafting of acrylamide on the CAAMC backbone. The above data show that cellulose was successfully grafted with acrylamide and acrylic acid, and MNC was successfully combined with the superabsorbent resin network via a chemical bond.

Figure 5A shows the SEM observation of the surface morphology of the nano-CaCO_3_; from the figure, we can see that the nano-CaCO_3_ particles obtained by solid-phase reaction are relatively uniform and spherical. In the literature, nano-CaCO_3_ was prepared by other mechanical processes, and the particle size was not uniform [57]. This is mainly due to the addition of triethanolamine and ethanolamine, which prevents the continued growth of calcium carbonate crystals and makes spherical particles easier during the ball milling process. This shows that nano-CaCO_3_ with uniform morphology is easier to prepare by the solid-phase reaction of ball milling, and the reaction process is mild and easy for industrial production. Figure 5B shows the SEM observation of the surface morphology of the modified nano-CaCO_3_; after nano-CaCO_3_ was modified by KH550, nano-CaCO_3_ has a little bonding phenomenon, which was similar to the previous research results [45], mainly because the molecular chains of KH-550 that connect with the surface of nano-CaCO_3_ particles produced mutual exclusion and a steric hindrance effect, and, thus, the surface free energy is reduced correspondingly and the agglomeration is controlled effectively [56]. From the particle size distribution results (as shown in Appendix A), the particle size distribution of unmodified nano-CaCO_3_ is uniform (basically around 25 nm), the particle size of MNC is about 70 nm, and the distribution range of particle size is between 50 and 90 nm, which is basically consistent with the SEM results. Figure 5C,D show the suspension of nono-CaCO_3_ and MNC in water. MNC maintains a better dispersed state in water; after being dispersed in the aqueous solution for 50 min, it still maintained a good dispersion. The unmodified nano-CaCO_3_ showed obvious sedimentation after 50 min, and the literature also reported a similar situation [58,59]. This is because the surface of the MNC is rich in organic groups, which makes it have good dispersibility in aqueous solution. On the other hand, it shows that nano-CaCO_3_ was successfully modified by KH550.

Figure 6 shows the SEM observation of the surface morphology of the CAAMC. Figure 6A shows that the surface of freeze-dried CAAMC had multiple pores and an obvious reticular structure. As shown in Figure 6B,C, the crosslinking between networks in multiple directions can be observed by magnifying the network structure, thus forming a 3D network structure. The cross-sectional view (Figure 6D) of the CAAMC shows the existence of a network formed by crosslinking inside the resin. These well-developed network structures allow water molecules to enter the superabsorbent resin easily and increase the water storage space of resin. Figure 6E shows that the surface of normal drying CAAMC was coarse and irregular, which gives CAAMC a larger specific surface area, increases the contact area between CAAMC and water molecules, and speeds up the rate of water absorption. Figure 6F shows the EDS Ca element mapping of the CAAMC, which clearly shows the uniform distribution of MNC in CAAMC.

Appendix A shows photos of a drop of water on bagasse, BC, and CAAMC when placed. From this, we obtain the instantaneous contact angles of water droplets with different material surfaces. The contact angle between the water droplet and the bagasse retrieved from the sugar mill is approximately 63°. When sugarcane bagasse is pre-treated, the contact angle between the BC and water droplets is about 54°, and the contact angle between the CAAMC and the water droplets was the smallest, which was about 47°. This is an indication that the superabsorbent resin prepared by modifying bagasse cellulose has much higher hydrophilicity than bagasse itself, which is very favorable for water absorption. In addition, according to Young-Laplace [60], the smaller the contact angle, the lower the surface tension, making it easier for water droplets to spread out on the surface of the superabsorbent resin, which makes the water absorption speed faster.

### 3.2. Polymerization Mechanism of CAAMC

Figure 7 shows the graft polymerization mechanism of MNC, AM, and AA with bagasse cellulose. First, APS was decomposed to form sulfate anion radicals under heating conditions. The radicals extracted hydrogen from the hydroxyl groups in cellulose, resulting in the formation of more active groups [61]. These active groups became reaction sites, where the monomers became acceptors; then, the monomer molecules themselves became free radical donors to the nearby molecules, resulting in the growth of polymeric chains [48]. At the same time, sulfate anion radicals attack the amino group on MNC and extract hydrogen from the amino groups, resulting in the formation of more active groups [62]. Then, the position where modified MNC loses hydrogen recurs and reacts with the monomers acrylic acid and acrylamide. Finally, under the action of the crosslinking agent MBA, MNC grafted with acrylic acid and acrylamide and cellulose grafted with acrylic acid are crosslinked to form superabsorbent CAAMC with a 3D network structure.

### 3.3. Effect of MNC Content on the Properties of CAAMC

#### 3.3.1. Effect of MNC Content on the Water Absorbency

The addition of inorganic materials impacts the structure and properties of SAR materials [34,63]. Hence, the water absorbency, salt absorption rate, reswelling performance, water retention effect, and mechanical strength of the composite SAR were studied after the addition of different modified MNC contents of 0–0.45 g (relative to 0–6 wt% of the total mass of the monomer) to understand the effect of MNC on the properties of the superabsorbent resin.

A comparison of the maximum water absorbency on different biopolymer-based SARs is summarized in Table 1. A maximum water absorbency in deionized water of 712 g/g and a maximum water absorbency in 0.9 wt% NaCl solution of 72 g/g for CAAMC were obtained in this study, which were higher than those in previous works (as shown in Table 1), indicating CAAMC exhibited a good water absorbency in 0.9 wt% NaCl solution compared with those of other SARs, which was probably because the addition of modified calcium carbonate effectively improves the salt resistance of CAAMC and forms a good network structure, so that CAAMC has a better water absorbency in 0.9 wt% NaCl solution.

Figure 8A shows that the water absorbency of CAAMC increased with the weight of MNC until 0.2 g (3.33 wt%), where the water absorbency gradually decreased. This is because when more MNC is added, the amino group on the surface will participate in the polymerization as a crosslinking component, which is crosslinked with organic components to produce a 3D network structure that can retain more water. When the MNC content becomes too high, the crosslinking density increases and the network becomes difficult to expand after absorbing much water, making it difficult to absorb more water. Moreover, the MNC content has a positive effect on the absorption of 0.9 wt% NaCl solution by the resin.

Figure 8B shows that the addition of MNC increased the absorption capacity of the resin 0.9 wt% NaCl solution and significantly increased the absorption rate. This is because MNC, as the crosslinking component, makes the interior of the superabsorbent resin form a certain spatial result when it starts to absorb water. It not only relies on the COO- group in the superabsorbent resin to repel each other but also increases the network space to accommodate more water, thus reducing the effect of ions on water absorption [37,70]. However, the crosslinking density becomes extremely high when the MNC content is excessive, decreasing the water absorbency.

#### 3.3.2. Effect of MNC Content on the Reswelling Water Absorbency of CAAMC

To evaluate effect of MNC content on the reswelling capability of CAAMC, reswelling capability studies were performed on CAAMC containing different MNC contents through measuring their swelling capacity loss during sequential swelling/drying cycles. Figure 9A shows that the addition of MNC improves the reswelling capability of the superabsorbent resin. After twice reswelling the superabsorbent resin without MNC, the resin disperses in the water when it reabsorbs water and can no longer be used; moreover, after the second swelling/drying cycle, the water absorbency rate is only 60% of the first water absorbency. The reswelling capability of the superabsorbent resin significantly improves by adding MNC, and the water absorbency can be >90% of the initial water absorbency when it undergoes a swelling/drying cycle three times. This behavior is observed because the superabsorbent resin forms a 3D network structure after adding MNC, and there is a close relationship between the polymer chain and the chain, thus improving the stability of the resin.

#### 3.3.3. Effect of MNC Content on Water Retention of CAAMC at Different Temperatures

The water retention capacity of the superabsorbent resin after water absorption and saturation is a vital property of the superabsorbent resin because this property affects the superabsorbent service life and application prospects of resin. Figure 9B–F show the water retention capacity of the superabsorbent resin at different temperatures after adding different amounts of MNC. Figure 9B shows the water retention effect of CAAMC at 20 °C. CAAMC has a good water retention effect near room temperature, and the highest water retention rate of CAAMC reached more than 80% after 5 h, which is 20% higher than that of the SAR without MNC addition. The water retention rate of CAAMC gradually decreased with increasing temperature. When the SAR was left at 30 °C for 5 h (as shown in Figure 9C), the SAR without MNC lost almost all water, but CAAMC retained approximately 40% of the water. When the temperature continued to increase, the SAR without MNC lost water completely after 5 h, and CAAMC still retained a small portion of water molecules (as shown in Figure 9D–F).

In summary, MNC significantly improved the water retention capacity of CAAMC. The best water retention was achieved when the MNC was 3.33 wt% (compared to the total monomer mass). The water retention effect of the SAR varies with the amount of MNC added, because the addition of MNC affects the entire superabsorbent resin structure, thus affecting its water retention performance, and SEM images of different MNC added to superabsorbent resin confirmed this explanation.

#### 3.3.4. Effect of MNC Content on the Gel Strength of CAAMC after Water Absorption and Saturation

Figure 10 shows the compressive stress-strain curves of CAAMC with different MNC contents after water absorption and saturation. We observed that the gel strength after water absorption improves after the addition of MNC, the water absorbency of CAAMC prepared by adding 2.00 wt% MNC was 200 g/g higher than that of superabsorbent resin prepared without MNC, and the gel strength of CAAMC after water saturation was twice that of superabsorbent resin without MNC. Therefore, we believe that the addition of MNC can effectively improve the gel strength of the superabsorbent resin, and the mechanical strength increases with MNC content because the addition of MNC, as a crosslinking component, participates in polymer formation, increasing the number of intersection points in the crosslinking network and resulting in a denser network with an increase in the hydrogel’s stress support point. Furthermore, MNC, as an inorganic component, has better compression resistance, which considerably improves the mechanical properties of the hydrogel.

#### 3.3.5. Effect of MNC Content on the CAAMC Morphology

Figure 11A–E show the surface morphology of CAAMC after the addition of different contents of MNC. We observed that changes in MNC content significantly influenced the morphology of CAAMC. The surface of the superabsorbent resin was relatively smooth in the absence of MNC; however, when it was added, the morphology of CAAMC exhibited a more obvious network structure. Nevertheless, excessive amounts of MNC increased the density of CAAMC morphology, indicating the involvement of MNC addition in the polymerization process as a crosslinking component. Therefore, the CAAMC structure can be adjusted by varying the MNC content, affecting the properties of CAAMC. For example, the water absorbency of the SAR in deionization was 490 g/g when MNC was not added, and the water absorbency of CAAMC reached approximately 710 g/g when an appropriate amount of MNC was added.

#### 3.3.6. Water Absorbency of CAAMC in Different pH Solutions and Different Salt Solutions

CAAMC has carboxylic acids, carboxylic amides, and hydroxyl groups, which exist in most anionic superabsorbent materials. The ionic SAR demonstrated different water absorbency over a wide pH range. The maximum water absorbency of the SAR in the pH range of 1.0–13.0 was investigated to examine the sensitivity of CAAMC to pH. Different pH solutions were obtained by diluting 0.1 mol/L HCl and 0.1 mol/L NaOH with distilled water [39]. The results (Figure 11F) showed that the water absorbency of CAAMC increased with pH from 1.0 to 6.0 and remained almost unchanged in the pH range of 6.0 to 9.0, because most of the carboxyl groups in CAAMC are protonated in an acidic solution (pH < 5), and the hydrogen bond interaction between carboxylates is enhanced, increasing the physical crosslinking density and reducing the equilibrium swelling ability of CAAMC. Furthermore, the electrostatic repulsion between carboxylate ions is weakened because of the protonation of carboxylates; thus, the water absorbency of superabsorbent resin is reduced. At higher pH values (5 < pH < 10), the electrostatic repulsion between carboxylate anions is enhanced because of the dissociation of carboxylic acid groups (-COOH → -COO-^+^H^+^). This phenomenon leads to additional expansion of the hydrogel network, thus improving the swelling capacity. The decrease in water absorbency in a highly alkaline solution (pH > 9) is attributed to the “charge shielding effect” of excess Na^+^ in the swelling medium, which shields carboxylates and prevents anion–anion repulsion [71].

Figure 11G–I show the influence of NaCl, CaCl_2_, and FeCl_3_ salt solutions on the water absorbency of CAAMC. The water absorbency decreased with an increase in the concentration of the three salt solutions and the ionic strength of the solution, which is attributed to the shielding of anionic hydrophilic groups by the counterions Na^+^, Ca^2+^, and Fe^3+^ and weakens the repulsive force between these anionic groups. Furthermore, we observed that the hydrogel adsorbed a large number of Fe^3+^ ions when CAAMC absorbed water and swelled in the FeCl_3_ solution (Figure 11K). Note that CAAMC exhibited a transparent gel-like appearance after absorbing water and saturating in deionized water (as shown in Figure 11J), whereas it exhibited a reddish-brown color in FeCl_3_ solution, indicating that a large amount of Fe^3+^ was adsorbed on CAAMC. The difficulty of the superabsorbent resin absorbing a large amount of water in the high salt solution is because the super-absorbing group on the surface of the superabsorbent resin complexes has a high salt ion content, which leads to the weakening of the mutual repulsive force between anions and the inability to open the internal network structure, thus resulting in a significant decrease in water absorbency [72,73].

Furthermore, the superabsorbent resin synthesized by adding a small amount of modified CaCO_3_ has a higher liquid absorption capacity in high-pH and low-pH solutions than the absorbent without modified calcium carbonate. Moreover, the swelling ability of the superabsorbent resin synthesized by adding a small amount of MNC in different salt solutions was higher than that of the superabsorbent resin without MNC. This is because the small amount of MNC added forms a 3D network structure with organic substances, such as cellulose, acrylic acid, and acrylamide, such that the resin itself has a large number of pore structures when the superabsorbent resin cannot expand to absorb more water molecules.

### 3.4. Water Retention in Soil

Figure 12 shows the water retention capacity of soils with different contents of CAAMC. As shown in Figure 12, with the increase of CAAMC content, the water retention effect of soil was significantly improved, and the water content of soil without CAAMC after complete wetting was also significantly lower than that of soil with CAAMC. With the extension of storage time, the water content of all soils decreased significantly. After 6 days of storage, the soil water content without CAAMC was only 13 wt%, while the soil water content with 2 wt% CAAMC still reached 120 wt%. This indicated that CAAMC could effectively improve soil water retention capacity.

### 3.5. Simulation Calculation

Figure 13A,B show the simulation of the adsorption process of water molecules by individual chains of SAR with different functional groups. We observed that the single chain of SAR containing the carboxyl group slacked when adsorbing water molecules, and the branched chains formed after the grafting of acrylic acid changed from a curved state to a straight chain. Moreover, the carboxyl group was distant from the SAR skeleton (as shown in Figure 13A), increasing the volume of the superabsorbent resin and absorbing additional water molecules. However, when the single chain of SAR containing an amine group adsorbs water molecules, the entire structure does not slack and maintains the original shape (as shown in Figure 13B). At the same time, we found that CAAMC has different water absorption processes in deionized water and 0.9 wt% NaCl solution (as shown in Appendix A). This is mainly because, in the salt solution, the water absorption rate of CAAMC is not only affected by the volume swelling rate but also by salt ions, which reduce the adsorption capacity of the functional groups contained in CAAMC to water molecules.

The adsorption energies of water molecules on different functional groups were calculated to examine the contribution of the amino and carboxyl groups to water absorption for the entire procedure. Water molecules form four types of hydrogen bonds with carboxyl and amine groups: the hydrogen atom on the amine group and the oxygen atom on the water molecule; the nitrogen atom on the amine group and hydrogen atom on the water molecule; the oxygen atom on the carboxyl group and hydrogen atom on the water molecule; and the hydrogen atom on the carboxyl group and oxygen atom on the water molecule. As shown in Figure 13C–P, different adsorption models are established based on the types of hydrogen bonds and adsorption modes.

When the water molecule is adsorbed on the functional group of the superabsorbent resin, it forms a hydrogen bond with the functional group, thus improving the adsorption effect of the superabsorbent resin on water molecules. Table 2 shows the adsorption energies of different functional groups to water molecules under different configurations. The maximum energy released by water molecules that adsorb and form a hydrogen bond with the carboxyl group is −2.240 kJ/mol, whereas the maximum energy released by the water molecule after adsorbing and forming a hydrogen bond with the amine group is −2.413 kJ/mol. In different adsorption configurations, the energy released by the water molecule after forming the hydrogen bond is higher than that of the carboxyl group, indicating that the amine group contributes more to the adsorption of water molecules in the water absorption process. The results show the importance of using two monomers to modify sugarcane cellulose. The SAR grafted with acrylamide exhibited stronger water absorption, and the SAR grafted with acrylic acid increased the volume of the superabsorbent resin in the water absorption process, thus absorbing additional water molecules. Therefore, the superabsorbent resin modified by the two monomers simultaneously showed good water absorbency.

## 4. Conclusions

We have fabricated a DC Bagasse cellulose-AA-AM copolymer composite SAR by double crosslinking. MNC and MBA as a double crosslinker contribute to form a stable network. The CAAMC exhibits a 3D network structure, and the maximum water absorbency in deionized water is 712 g/g and in 0.9% NaCl solution is up to 72 g/g. The results of FT-IR and XPS showed that nano-CaCO_3_ was modified with silane coupling agent, and the MNC were combined with inorganic components via chemical bonding. The structure of CAAMC can be changed by adjusting the content of MNC, and the addition of MNC improves the mechanical properties, reusability, and water retention of CAAMC. Adding a small amount of CAAMC to the soil can effectively improve the water retention capacity of the soil, indicating that CAAMC has great application potential in soil water retention. In addition, the results of MS calculations showed that the amine group in the superabsorbent resin exhibits a stronger adsorption ability to water molecules, whereas the presence of the carboxyl group causes the superabsorbent resin to have a larger volume after water absorption. This study provides a novel procedure for preparing SAR from inorganic composite organics, and the theoretical simulations of the absorption process provide theoretical support for the further study of superabsorbent resin.

## Figures and Tables

**Figure 1 nanomaterials-12-01459-f001:**
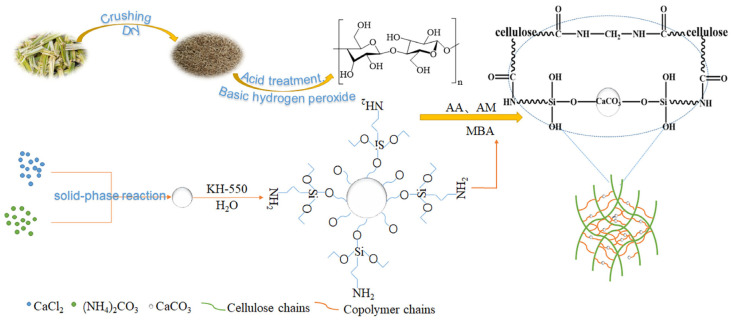
Preparation process of composite SAR CAAMC.

**Figure 2 nanomaterials-12-01459-f002:**
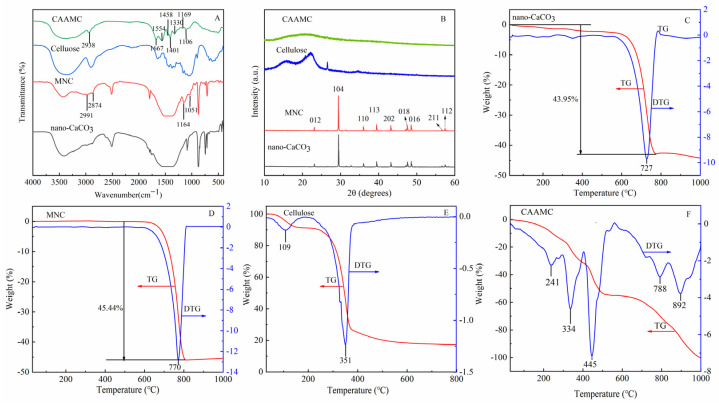
FT-IR spectra (**A**) and XRD patterns (**B**) of nano-CaCO_3_, MNC, cellulose, and CAAMC; TG-DTG curves of nano-CaCO_3_ (**C**), MNC (**D**), cellulose (**E**), and CAAMC (**F**).

**Figure 3 nanomaterials-12-01459-f003:**
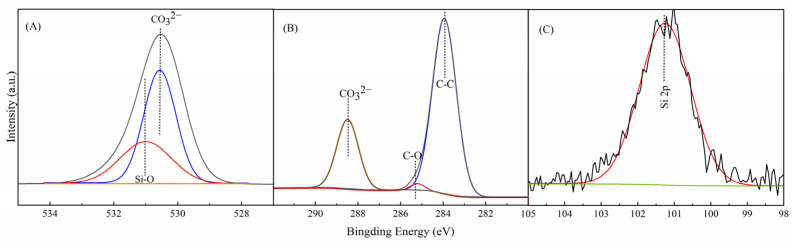
XPS profiles of O 1s (**A**), C 1s (**B**), and Si 2p (**C**) for MNC.

**Figure 4 nanomaterials-12-01459-f004:**
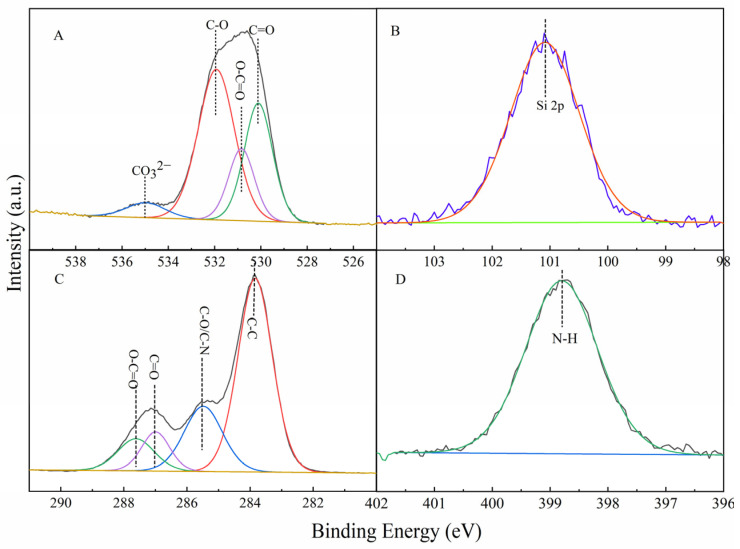
XPS profiles of O 1s (**A**), Si 2p (**B**), C 1s (**C**), and N 1s (**D**) peaks for CAAMC.

**Figure 5 nanomaterials-12-01459-f005:**
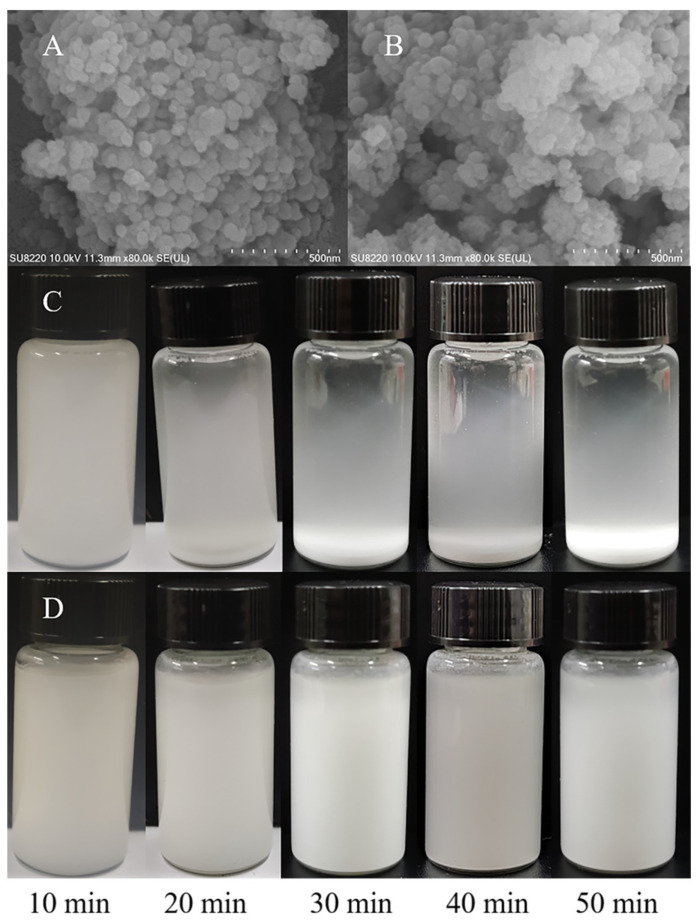
SEM images of nano-CaCO_3_ (**A**) and MNC (**B**), and the dispersion of nano-CaCO_3_ (**C**) and MNC (**D**) in aqueous solution.

**Figure 6 nanomaterials-12-01459-f006:**
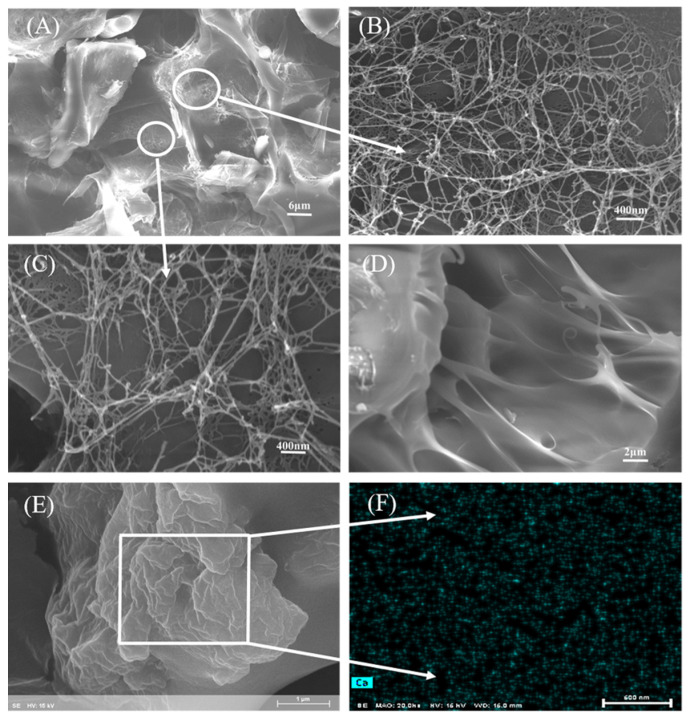
SEM images of the freeze-dried CAAMC surface (**A**), magnified small holes on the CAAMC network wall (**B**,**C**), CAAMC cross-sectional view (**D**), normal drying CAAMC surface (**E**), and the corresponding EDS Ca element mapping of the CAAMC (**F**).

**Figure 7 nanomaterials-12-01459-f007:**
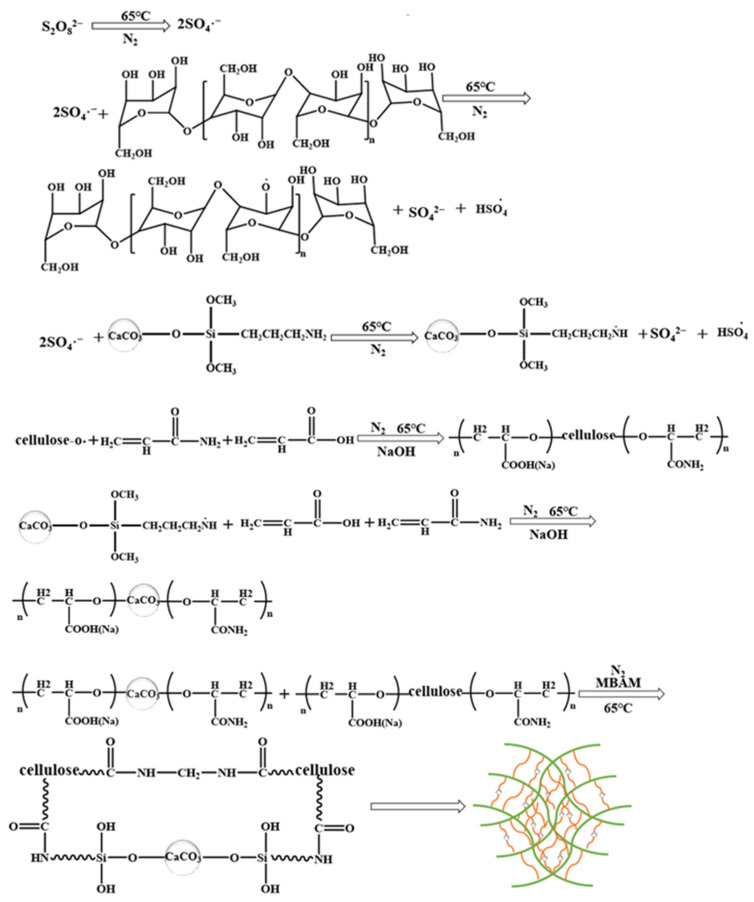
The suggested grafting mechanism for MNC, AA, AM, and BC.

**Figure 8 nanomaterials-12-01459-f008:**
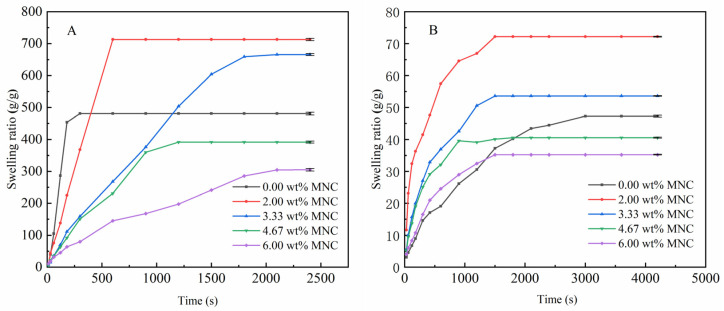
Effect of MNC content on the water absorbency of CAAMC in deionized water (**A**) and 0.9 wt% NaCl solution (**B**).

**Figure 9 nanomaterials-12-01459-f009:**
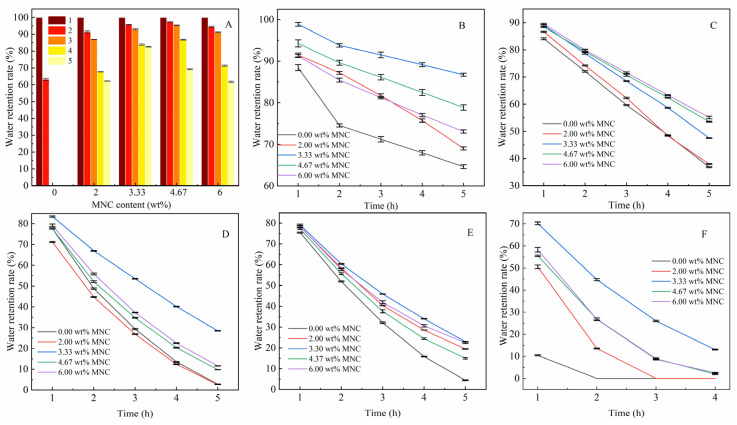
Swelling/drying cyclic water absorbency (**A**) and water retention rate at 20 °C (**B**), 30 °C (**C**), 40 °C (**D**), 50 °C (**E**), and 60 °C (**F**) of CAAMC.

**Figure 10 nanomaterials-12-01459-f010:**
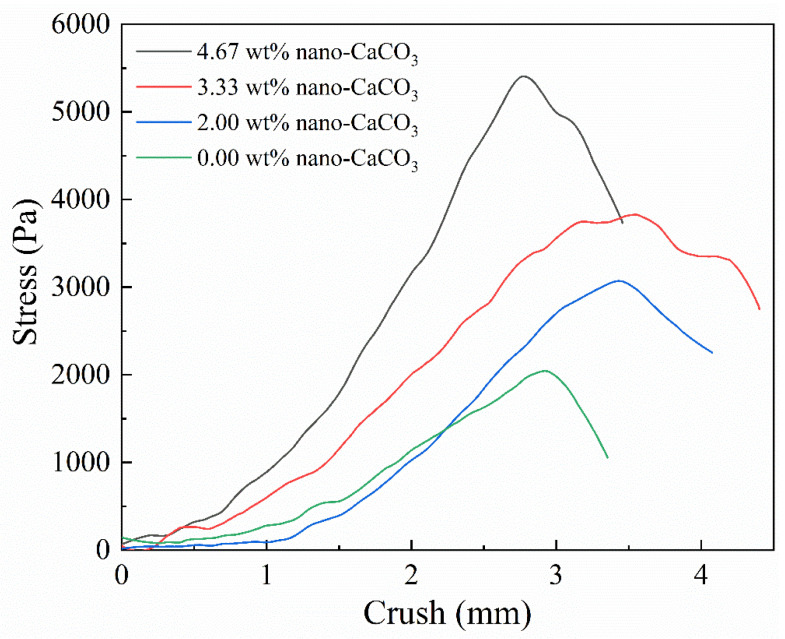
The gel strength of CAAMC after swelling equilibrium.

**Figure 11 nanomaterials-12-01459-f011:**
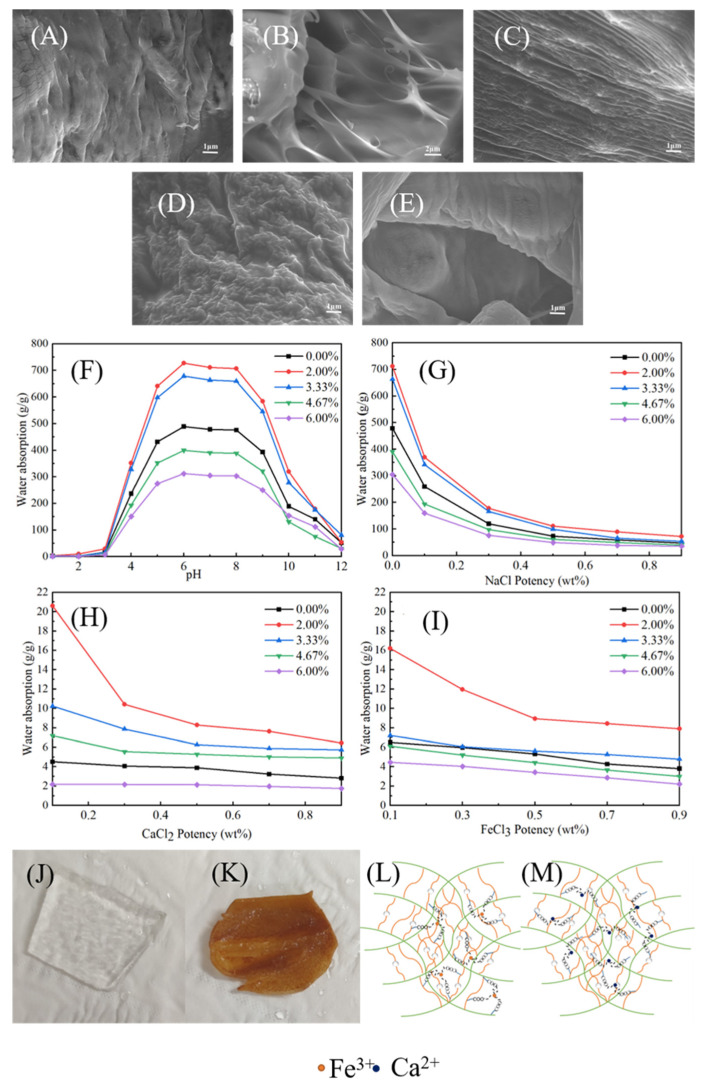
SEM images of CAAMC without MNC (**A**) and with 2 wt% MNC (**B**), 3.33 wt% MNC (**C**), 4.67 wt% MNC (**D**), and 6 wt% MNC (**E**); effects of solution pH on the water absorbency of CAAMC with different modified calcium carbonate contents (**F**), the water absorbency of CAAMC with different MNC contents in NaCl (**G**), CaCl_2_ (**H**), and FeCl_3_ (**I**) aqueous solutions; appearance of CAAMC after swelling equilibrium in deionized water (**J**) and the FeCl_3_ solution (**K**); structural models of CAAMC at salt solutions FeCl_3_ (**L**) and CaCl_2_ (**M**).

**Figure 12 nanomaterials-12-01459-f012:**
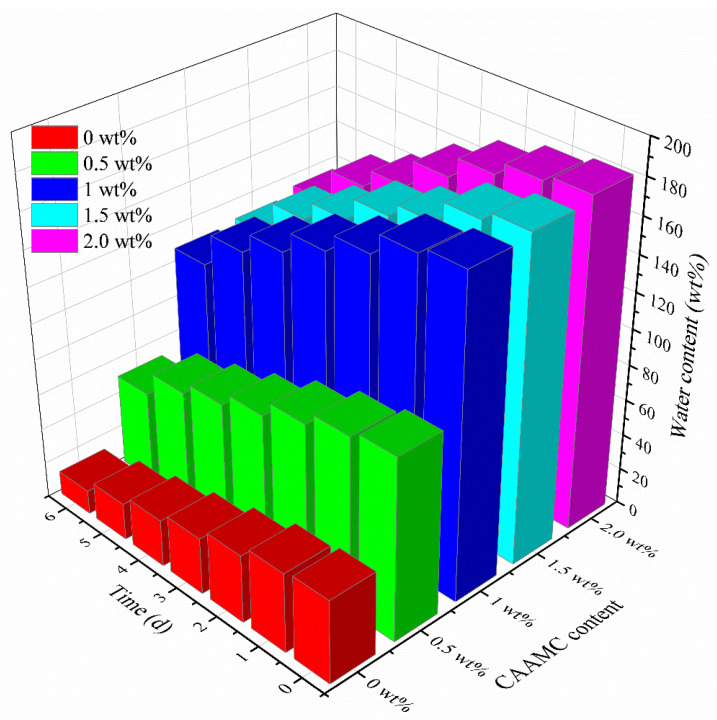
Water retention effect of soils with different CAAMC contents.

**Figure 13 nanomaterials-12-01459-f013:**
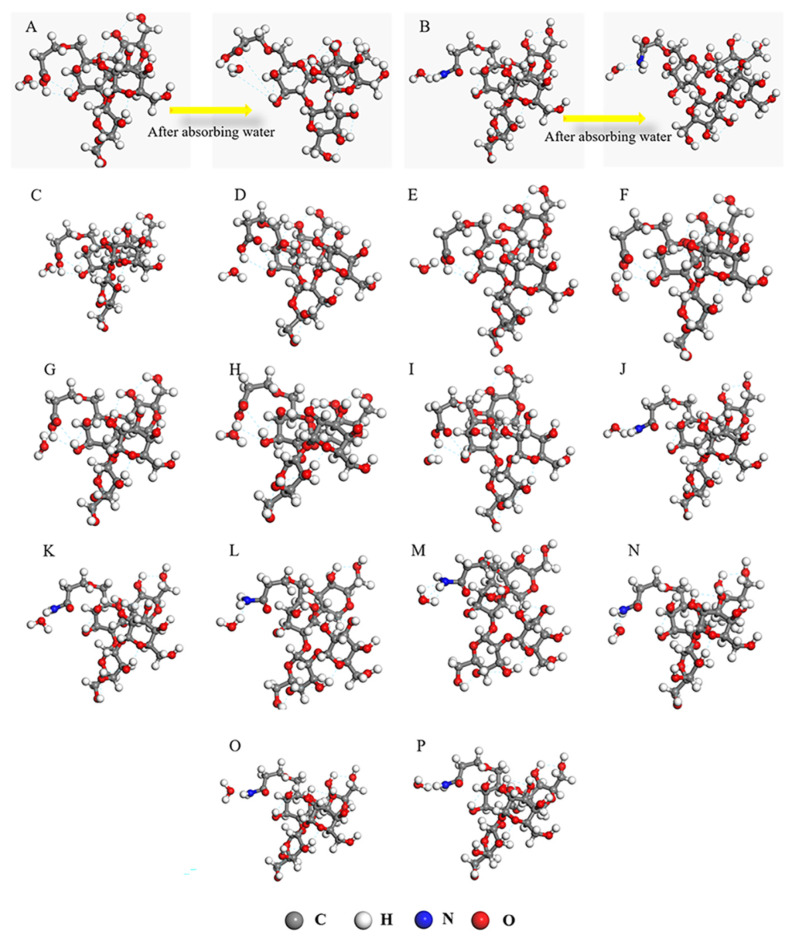
Structural changes in the adsorption of water molecules by carboxyl (**A**) and amino (**B**) groups in CAAMC branched chains and different configurations of adsorbed water molecules (**C–P**).

**Table 1 nanomaterials-12-01459-t001:** Comparison of water absorbency of CAAMC with that of other SARs.

Raw Materials	Ratio of Monomers (Monomer:Natural Polymer Material)	Water Absorbency in Deionized Water (g/g)	Water Absorbency in 0.9 wt% NaCl (g/g)	Ref.
Cellulose, AA, AM	10:1	240	-	[64]
Carboxymethyl cellulose, PAA, Graphene oxide	25:3	750	85	[65]
Cellulose, 1,2,3,4-butanetetracarboxylic dianhydride	3:1	987	-	[66]
cotton stalk, bentonite, polyvinylpyrrolidone	9:1	1018	71	[67]
Corn straw cellulose, ammonium polyphosphate, AA	3:1	303	-	[68]
Starch, AA, AM, polyvinyl alcohol, cellulose nanocrystals	17.5:1	922	52	[12]
(2-pyridyl) acetyl chitosan chloride, AA, AM	10:1	615	44	[48]
Lignin, PVA	20:1	456	-	[69]
Bagasse cellulose, AA, AM, CaCO_3_	8:1	712	72	This work

**Table 2 nanomaterials-12-01459-t002:** Energy changes before and after the adsorption of water molecules in different configurations.

Configuration	Pre-Adsorption Energy (kJ/mol)	Post-Adsorption Energy (kJ/mol)	Adsorption Energy (kJ/mol)
a	32.613	31.265	−1.348
b	32.613	31.443	−1.170
c	32.613	31.368	−1.245
d	32.613	31.062	−1.551
e	32.613	31.512	−1.101
f	32.613	30.373	−2.240
g	32.613	31.847	−0.766
h	33.010	32.452	−0.558
i	33.010	31.200	−1.81
j	33.010	30.918	−2.092
k	33.010	31.576	−1.434
l	33.010	31.992	−1.018
m	33.010	30.597	−2.413
n	33.010	32.056	−0.954

## Data Availability

The data presented in this study are available on request from the corresponding author.

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
