# Peer review of "Bagasse Cellulose Composite Superabsorbent Material with Double-Crosslinking Network Using Chemical Modified Nano-CaCO3 Reinforcing Strategy"

_nanomaterials, 2022, doi:10.3390/nano12091459_

Round 1

Reviewer 1 Report

Some of the suggestions are accepted, leading to certain improvements in the manuscript. 

Reviewer 2 Report

The authors have addressed the reviewer's comments well

This manuscript is a resubmission of an earlier submission. The following is a list of the peer review reports and author responses from that submission.

Round 1

Reviewer 1 Report

Interesting results and novelty work. A paper focuses on Bagasse cellulose composite superabsorbent material with double-crosslinking network using chemical modified nano-CaCO3 reinforcing strategy. Though the intention of the authors is highly commendable, there is lot of problems particularly in the presentation throughout the manuscript. Besides there are many grammatical mistakes throughout the manuscript, particularly in respect of use of singular and plural with the subject or verb. In view of the above comments, whole manuscript should be properly written to make it acceptable by Nanomaterials. I highly recommended this article to be accepted and published in the revised version.

 Abstract:

The abstract given here starts without any background for the present work. Of course, it contains brief details about experimental aspects and the obtained results. However this abstract does not follow the norm of an abstract, which should state briefly:

  1. The purpose of the study undertaken, what are you trying to solve
  2. brief mention of experimental aspects (without using abbreviations)
  3. highlights of the results numerically
  4. Important conclusions based on the obtained results
  5. Potential applications

Therefore, it is suggested that the Abstract to be modified as per the suggestions given above.

 Introduction

Introduction section is long with a many references based on the literature survey conducted by the authors. This is very good. However, this lacks in proper presentation of literature survey, which should have been systematic whereby existing scientific gaps should have been brought out. This should have given justification for the present study, which should be followed by the objectives of this study. In fact there is large amount of literature available on the characterization of superabsorbent resin from cellulose. Similarly, a large number of methods to obtain these materials have been used mentioning their advantages and limitations. None of these have been brought out in this study whereby the authors have not justified why they have chosen the method they have used in their study.  It should be noted that normally 'Introduction' should give brief background through literature survey for the study citing previous published work where-by scientific gaps that exist should be brought out. This would have led to justification for the present study.  It is therefore suggested that ‘Introduction Section’ should be revised as suggested above because this Section is an important one from the point of view of taking up the present study.

In my opinion the paper will have good merit if such applications can be demonstrated and reported. Can you give some example?

Relevant article on natural fiber and cellulose should be cited such:

Polymers, 8(6), 1–28. https://doi.org/10.3390/polym10060623

Cellulose, 24(3), 1171–1197. https://doi.org/10.1007/s10570-017-1194-0

Carbohydrate Polymers, 83(2), 953–958. https://doi.org/10.1016/j.carbpol.2010.09.005

Carbohydrate Polymers, 119, 202–209. https://doi.org/10.1016/j.carbpol.2014.11.041

Polym Test 2020;91:106751. https://doi.org/10.1016/j.polymertesting.2020.106751.

Curr Anal Chem 2020;16:500–3. https://doi.org/10.2174/157341101605200603095311.

Polymers (Basel) 2021;13. https://doi.org/10.3390/polym13030471. 

Bagasse, a by-product of the sugar cane industry containing 40–50 wt% cellulose, 25–35 wt% hemicellulose, and 18–24 wt% lignin, contain many substances hy-droxyl and phenolic groups. Please cite this sentence.

Materials and Methods:

Normally, this section should have two main subsections. The first one is Materials which should give details of all materials used in the study, where from they were procured, known characteristics, if available (for e.g. bagasse, CaCl2, (NH4)2CO3, 3-aminopropyltrieth- 98 oxysilane, potassium persulfate, acrylamide, N, N’-methylene bisacrylamide, where do you get it, what is the purity of the chemical and etc.).

The second subsection should be Methods, where methodologies used in the study should be given in a systematic way using sub section with numbers for each of the properties. First the processing or preparation aspects of the final material should be given followed by the characterization of prepared materials including preparation of samples for any specific property or morphology studies should be presented in a systematic way. Here one should also clearly mention the number of samples used, any standards followed for variety of properties, make and model of the instruments used for characterization, their accuracy and experimental conditions used, etc.

It should be known to the authors when one publishes any scientific paper, the results presented therein should be such they should be reproducible by any other person when the experiment is repeated using the same materials. In the present paper, it would be difficult for any other person to repeat the experiments because the chosen materials do not have any pre-history, which is required for other researchers to carryout experiments to check the possible reproducibility of the procedure adopted by these authors.

Some of the paragraph should be under results and discussion and if it is already there then this becomes repetition and hence can be deleted. Secondly, this Section is methods and hence only results should be mentioned and then it should be discussed preferably comparing it with earlier reported similar results by other researchers.

Results & Discussion

Well written and easy for the reader to understand what the authors have conveyed.

Some of the paragraph should be under Methods and if it is already there then this becomes repettion and hence can be deleted. Secondly, this Section is Results & Discussion and hence only results should be mentioned and then it should be discussed preferably comparing it with earlier reported similar results by other researchers.

Throughout the manuscript, there are less comparison had been done with other published journal. Therefore, please support your statements with other researcher’s work in the section result and discussion. It should be discussed preferably comparing it with earlier reported similar results by other researchers.

Please Figure 4, 8 a-e.

How many sample did for each experiment? Please do ANNOVA test and standard deviation for all data collected and presented.

Conclusions

Conclusions given here are do not reflect what had been achieved including many speculations. It is too long and should be in 1 paragraph. Hence these need to be suitably modified. It may be remembered that this Section forms a summary of all the major observations/ results obtained. Accordingly, here presentation should consist of the main Results or the observations of the study in short sentences probably with bullet points. This should stand alone or form a subsection of a Discussion or Results Section. Hence better to rewrite this Section based on the comments given in the whole text.

General Comments:

The paper though contains some interesting results and novelty work, it lacks in its proper presentation in the whole manuscript. Of course there is need for better language throughout the manuscript. It is suggested that the authors should take the help of native English speaking person to take care of this problem. In view of these, the paper is highly recommended and should be accepted for publication in the revised form. It is suggested that the authors should revise the paper in the light of above comments/suggestions.

Reviewer 2 Report

The authors performed significant effort and a lot of work is done within the subject that is worth of investigation, Unfortunately, based on the presented results and conclusions, in my opinion the submitted manuscript cannot be published in the appreciated and prestigious Journal “Nanomaterials”,  The main reasons for my decision for this recommendation is summarized in the “Comments and Suggestions for Authors
”.

Comments and Suggestions for Authors

  1. An overview of the state-of-the-art is not adequate presented in regard to the subject of the work. The comprehensive literature overview about the synthesis of bagasse-based SARs, as said by the authors, actually superabsorbent, is not presented. The same is for the use and effects of (nano)CaCO3 and modified (nano)CaCO3 in superabsorbent.
  2. Specific comments in regard to the Introduction:  
    • p/1. Line 32-37 “However, in the past the raw materials of SAR for commercial applications mainly come from poorly degradable petrochemical products such as polyacrylamide, polyacrylic acid [7,8] …. So the application of SAR in personal hygiene products and soil water-holding agents was limited “ – it is not wholly true. On the other side, actually, the SAR that has been investigated in this work and recommended due to its extraordinarily is significantly, in fact, based on polyacrylamide and polyacrylic acid.
    • Line -40-41 “Constructing a double crosslinking bi- mass-based-inorganic composite superabsorbent is an important method for solving the above problems and reducing carbon emissions” –should be approved.  
    • Line 70: “The mechanism of simulating the swelling process plays an important guide in the construction of superabsorbent resins.” – inappropriate phrase construction and not true. Actually, the model of swelling kinetics should be more appropriate.
    • Swelling kinetics: There are significant works in literature dealing with other kinetics model. Mandatory e take a look and involve in the next iteration of the manuscript !.
  3. The physico-chemical properties of the used bagase-cellulose (nano)CaCO3 and modified (nano)CaCO3 are not given. Besides the others, for bagase -cellulose it is mandatory to provide : i.e, Celulose %, type of crystalinity , etc and CaCO3 , particle size, particle size distribution,  purity, degree of modification, etc.
  4. The synthesis procedure should be described providing all necessary details including the amounts (ratios) of all the used materials. The description of procedure for SAR’s samples drying is problematic. is this procedure using through the investigation? Are the samples dry? (see TG/ DSC results!)? What’s about lyophilized samples? The authors do not provide information about the yield of the synthesized products and their purity.
  5. Water retention experimental procedure is not described.
  6. The configurations based on which are calculated adsorption energies (∆E) (eq 2) are not given.
  7. The FT-IR spectra’s assignation is poor and it is not supported with literature data.
  8. The XRD patterns of CAAMC do not exhibit any of the characteristic patterns for bagasse cellulose, nano-CaCO3 or MNC. It should be explained why. Literature data on XRD analyses for AA and AAM networks should be given and discussed the obtained results.  
  9. Based on the FT-IR spectra and their analyses it is not likely that it should be a valid proof for the suggested model of crosslinking and especially for double -crosslinking. The TG/ DSC curves also do not imply on additionally crosslinking.
  10. The TG / DSC curves reveals significant amount of water in sample. Be precise which sample is presented in Fig 2c. The TG process is not recorded until constant residual.
  11. The suggested mechanism of polymerization is little likely. There is no clear facts that can confirm the sated.
  12. There is not given explanation for the obtained values for adsorption in water and 0.9% NaCl.   
  13. “Table 1 Comparison of water absorbency on CAAMC with that on other SAR”, page 10, is not appropriate for R&D. It should be somewhat informative in certain context but it is necessary to know precise data about relevant structural properties of the SAP materials (monomers, ratio of monomers, crosslink density, inorganic particles, particle size, crystallinity type .. ) to make comparison and proper conclusions.
  14. Based on the presented results it is not obvious that significant improvement in mechanical properties is achieved. ( fig 7)
  15. Fig 8. Please explain the appearance of CAAMC after swelling equilibrium in deionized water (J) -transparent ? on my long lasting experimental experience it is litle likely in the cases when  contains significant amount of bagasse cellulose and CaCO3,  and it is in contrast with  images in Fig Fig 4 c and d.
  16. There are a lot of typo errors throughout the text. For example :

Line 100-101 : …from Macklin and  ? All chemical reagents utilized in this study were of analytical  grade  ….. with no further purification. ethical approval code. ?

Reviewer 3 Report

The paper “Bagasse cellulose composite superabsorbent material with double-crosslinking network using chemical modified nano-CaCO3 reinforcing strategy” presents an important topic since there is a growing interest towards bio-based superabsorbent materials. The paper is lengthy with so many results that the reader becomes quite exhausted. Some of the results presented were quite unexpected as they were not mentioned in the abstract / introduction / materials and methods prior to their presentation in the results section. Major revision is needed before the paper can be accepted for publication.

  • Regarding grammar, the paper could be barely understood at times, and therefore language revision is definitely needed. This applies especially to introduction and conclusions. Conclusions part contains quite exhaustive and long sentences (e.g. in lines 491-496).
  • Introduction had a curious structure: First the background was explained, then what was done in this study (lines 49-56), then back to the background. This makes the text rather confusing. Can you present the background first, summarizing the relevant literature and then explain what was done in this work?
  • 2 Characterization methods. This section is very scarce in details. There is no way that someone can repeat the measurements with this information. Please provide detailed information on how the characterization was done. In addition, XRD and EDS measurements were not described at all in the methods section but the results from these analyses were described.
  • Figure 3. Place the discussion related to the Figure before it, not after. Same with Fig. 7.
  • Line 256 discusses the mechanical strength of the superabsorbent resin. There were no description of the measurement in the methods section so I assumed that this was only a speculative statement. Only in section 3.3.4 it was revealed that some measurements of gel strength were actually made with material testing machine.
  • Line 307: “The maximum water absorbency in deionized water of 307 712 g/g and maximum water absorbency in 0.9 wt% NaCl solution of 72 g/g for CAAMC was obtained in this study. Which were higher than those in previous works (Table 1).” This is not true as there were higher values in the Table 1 to which the authors were referring to. If they are only comparing similar materials, CAAMC, then they should not include other type of materials in the Table or explain it better, i.e. which materials in Table 1 they are comparing.
  • 3.2 Performance of the material after multiple cycles. Recycling is not the correct term to be used here but only adds to the confusion. The recycling refers to the end-of-life of the material.
  • In general: many different results are grouped and presented in the same Figures, which makes it very difficult to follow the actual results and how the properties develop in the function of variables. This comment refers to both Fig. 2 and Fig 7. Consider splitting the results into more figures or leave some of the results out.
  • The authors refer to degradability of cellulose-based materials as superabsorbents. However, the bio-based materials in this case are heavily modified in this work. What kind of effect does the cross-linking of inorganic materials into bagasse cellulose have on the degradability of the material? Please elaborate.

Round 2

Reviewer 2 Report

Although authors gave responses to each of the comments, they are poorly implemented and insufficiently addressed in the revised manuscript. The revised manuscript is not significantly improved according to the given comments. The submitted manuscript is not acceptable for publication.

Author Response

Thank you very much for your re-review and suggestions. It is a pity that this revision did not satisfy you. We feel that your suggestion is very professional, which has a good guiding effect on our subsequent research. Due to the influence of time and the Corona Virus Disease 2019, the particle size and particle size distribution of the modified calcium carbonate have only been analyzed by SEM, and there is no further characterization analysis. But we believe that SEM analysis can understand the particle size distribution and particle size of nano-CaCO3 before and after modification to a certain extent, From the SEM analysis results, the particle size of nano-CaCO3 before and after modification is about 50 nm, and the particle size distribution is uniform. We regret that the degree of modification of nano-CaCO3 could not be characterized due to the influence of conditional testing.